# Rehabilitation Prognostic Factors following Hip Fractures Associated with Patient’s Pre-Fracture Mobility and Functional Ability: A Prospective Observation Study

**DOI:** 10.3390/life13081748

**Published:** 2023-08-15

**Authors:** Smaragda Koudouna, Dimitrios S. Evangelopoulos, Michail Sarantis, Efstathios Chronopoulos, Ismene A. Dontas, Spiridon Pneumaticos

**Affiliations:** 13rd Department of Orthopaedic Surgery, KAT Hospital, 145 61 Athens, Greece; koudouna.smaragda@gmail.com (S.K.); ds.evangelopoulos@gmail.com (D.S.E.); spirosgp@med.uoa.gr (S.P.); 2Department of Physiotherapy, KAT Hospital, 145 61 Athens, Greece; 34th Department of Orthopaedic Surgery, KAT Hospital, 145 61 Athens, Greece; 4Laboratory for Research of the Musculoskeletal System, National and Kapodistrian University, KAT Hospital, 145 61 Athens, Greece; stathi24@yahoo.gr (E.C.); idontas@med.uoa.gr (I.A.D.)

**Keywords:** CCI, HHS, hip fracture, prognostic factors, rehabilitation, SPBB, TUG

## Abstract

Low physical function is associated with poor outcomes in the elderly population suffering from hip fractures. The present study aims to evaluate the prognostic tools for predicting patient recovery after hip fractures and investigate the correlation between the pre-fracture motor and functional statuses. A prospective study was performed, including 80 patients suffering from hip fractures. Patient history, previous falls, the type of fracture and overall survival were evaluated. Patient-reported outcome measures (SF-36, EQ-5D/VAS, Charlson Comorbidity Index (CCI), Short Physical Performance Battery (SPPB), Timed Up and Go (TUG) and Harris Hip Score (HHS)) were monitored before hospital discharge at 6 weeks, and 3, 6 and 12 months postoperatively. Overall, 55% of patients experienced at least one fall, and 46% of them used crutches before the fracture. The average CCI score was 6.9. The SPPB score improved from 1.4 ± 1.3 (1 week) to 4.4 ± 2.1 (48 weeks). A one-year age increase, female sex, and prior history of falls lead to 0.1-, 0.92-, 0.56-fold lower SPPB scores, respectively, at 12 months. The HHS recorded the greatest improvement between 6 and 12 weeks (52.1 ± 14.6), whereas the TUG score continued to improve significantly from 139.1 ± 52.6 s (6 weeks) to 66.4 ± 54 s (48 weeks). The SPPB and performance test can be routinely used as a prognostic tool.

## 1. Introduction

Hip fragility fractures are one of the leading causes of morbidity in the elderly and are associated with an increased mortality of around 12–17% within the first year [1,2]. The majority of hip fractures occur in females aged above 80 years [3]. Epidemiological data vary between countries, but their frequency tends to increase due to increased life expectancy, with an incidence of 350/100,000 per year in developed countries [4], while it is expected that by 2050, 500,000 to 1 million elderly people will be affected [5]. The estimated annual cost in the US is approximately USD 10.3–15.2 billion [6].

Despite medical advances and surgical techniques, hip fractures continue to have a significant impact on elderly patients’ quality of life and increased socioeconomic consequences [7,8,9]. Half of the patients suffering from hip fractures regain independent gait and return to their previous mobility status, while only 25% fully recover [10]. Intensive hospital rehabilitation was significantly associated with a lower risk of mortality compared to rehabilitation in inpatient rehabilitation facilities (IRFs) and no rehabilitation [11]. Furthermore an interdisciplinary pathway in hip-fractured elderly patients could reduce in-hospital mortality, improve functional recovery and increase the probability of living alone at home, at 6 months [12].

Prognostic tools for clinical assessment of patients are widely used in research and clinical practice for the evaluation of therapeutic interventions. The present study aims to evaluate prognostic tools for predicting patient recovery following hip fractures and investigate the possible correlation with the pre-fracture motor and functional statuses. A scoring system of the various variables, which may independently affect the outcome, will be developed and validated to identify patients prone to the worst outcome.

## 2. Materials and Methods

The study was approved by KAT Hospital’s Scientific Council and Ethics Committee (6084/08-05-2018), as well as from the National and Kapodistrian University of Athens, School of Health Sciences, Department of Medicine (1819004332/04-10-2018), as an original doctoral study. Patient informed consent was received prior to participation in the study. The prospective cohort studied included 100 patients with hip fractures who were admitted to the emergency department between February 2019 and November 2020 and followed-up postoperatively for 12 months. All patients aged more than 50 years old and suffered from hip fracture after a fall from their own height, regardless of their medical history, were included in the study. Patients who suffered from multiple injuries with an ASA score of 4 or 5 were at their initial admission, hip fracture related with car accident or fall from stairs/height and patients lost before their final follow-up session (lost or dead) were excluded.

### 2.1. Reference Data

The major purpose of postoperative rehabilitation is to assist patients in returning to their pre-fracture condition. However, many patients are unable to achieve their pre-fracture level of autonomy. Age, gender, high body mass index (BMI), comorbidities, pre-fracture lack of autonomy and prolonged bed rest have been proposed as indirect prognostic mobility factors [13,14]. The quantification of preoperative and postoperative scores is essential for objective therapy evaluation. Preoperative activity can be measured and compared to postoperative-level activity to estimate the effect of postoperative rehabilitation on patients’ function. There are several useful scoring systems available for patients during their hospital stay and their follow-up period. 

The Charlson Comorbidity Index (CCI) predicts mortality, morbidity, and postoperative complications after a hip fracture [15,16,17]. Recent studies [18,19] on hip fracture patients undergoing intense inpatient rehabilitation showed that moderate to severe comorbidity and communication are prevalent and predict rehabilitation failure.

Health-Related Quality of Life (HRQoL) can be measured using a variety of generic questionnaires, including the Health Survey Questionnaire Short Form-36 (SF-36) and EuroQoL 5D/VAS (EQ-5D/VAS), which are frequently used in studies to assess health quality [20,21,22,23]. Following a hip fracture, patient quality of life deteriorates dramatically. The majority of patients recover within the first 6 months; however, they do not return to their pre-fracture health status. Comorbidities and complications were all strongly associated with health status and quality of life. Patients with total hip arthroplasty or hemiarthroplasty demonstrate better health statuses than those undergoing internal osteosynthesis [24].

Recurrent falls in the elderly are related with lower limb weakness. The Short Physical Performance Battery (SPPB) test and the Timed Up and Go (TUG) test are two tests routinely used to evaluate lower limb strength [25,26,27,28]. The SPPB comprises three items: standing balance, walking speed, and chair stands. Each item was evaluated on a scale from 0 (inability to complete) to 4 (best performance possible). The TUG evaluates the time it takes a person (in seconds) to stand up from an armed chair with a seat height of 46 cm, walk 3 m, turn around a cone, and return to the same chair. Participants are instructed to safely complete the test as quickly as possible, even with walking assistance. 

The Harris Hip Score (HHS) is a patient-reported questionnaire that includes assessments based on pain, function and range of motion [29]. One item evaluates the pain (0–44 points), seven items evaluate the patient’s functionality (0–47 points), one item evaluates the deformity (0–4 points) and two items evaluate their range of motion (0–5 points). The total points form a scale from 0 to 100. 

### 2.2. Procedures

Before surgery: Patients’ demographics (age, gender, and BMI), any comorbidities, and pre-fracture status were documented by completing questionnaires the day after admission (SF-36, EQ-5D/VAS, CCI, as well as a personal interview with the following questions: (1) Have you experienced any other falls? (2) Have you had any additional orthopedic procedures on your lower extremities? (3) Do you suffer from osteoporosis? (4) Do you have any eyesight issues? (5) Do you have Parkinson’s disease or another neurological condition? (6) Did you use a cane or a walker?) [24,25,26,27]. 

Furthermore, fracture type and surgical technique were documented as follows: (1) Intracapsular fracture: individual screws/percutaneous osteosynthesis, hemiarthroplasty, or total hip arthroplasty. (2) Intertrochanteric fracture: dynamic compression hip-screw or intramedullary nail. (3) Subtrochanteric fracture: long intramedullary nail with or without screws.

Finally, the type of anesthesia was noted as either regional or general.

After surgery: All patients underwent their mobility and functional performance tests 1 day before discharge using SPPB, TUG, and HHS [17,18,19,20,21,22,28,29]. In addition, overall length of hospital stay was documented as the time period between operation and discharge. 

Following hospital discharge and endpoint definition: Patients were followed-up for a period of 12 months. Patients who died at any point throughout the study were not included. During the 12-month follow-up, patients’ mobility and functional performance were evaluated using SPPB, TUG and HHS at 6 weeks, 3 months, 6 months, and 12 months. At the last follow-up, patients were also asked to complete the SF-36 and EQ-5D/VAS questionnaires to measure their quality of life. Finally, supplementary data were recorded, such as the stability of osteosynthesis, new falls, hospital readmissions, and overall survival.

### 2.3. Physiotherapy Protocol

The physiotherapy rehabilitation protocol included immediate mobilization, partial weight-bearing and muscle strengthening exercises. Patients were taught to reach sitting and standing positions from bed to chair, as well as walking with assistance. The muscle strengthening program included exercises to increase muscle strength and better control of the fractured limb: ankle pump exercises and quadricep exercises. The intensity of the exercise was proportionate to the patients’ discomfort and endurance.

Following discharge, patients were instructed to follow a physical therapy program for a period of three months with three 60 min weekly sessions focusing on enhancing safe mobility and improving muscle function. Strengthening exercises with progressively increasing resistance were administered to develop lower limb muscle strength as well as balance and endurance to improve performance of fundamental functional activities. Patients rehabilitation education protocol was focused in strengthening, balancing, and endurance exercises, particularly with functional activities of daily living (walking, sitting down on and standing up from a chair, and climbing and descending stairs).

## 3. Statistical Analysis

The continuous variable values are given using the number of participants (N), the mean value, standard deviation, median, and intra-quadratic range (in case the data do not follow a normal distribution). Frequencies (n) and corresponding percentages for categorical variables (percent) are utilized. The Kolmogorov–Smirnov test was used to determine the regularity of the measurement distribution. The Chi-square test, Fisher exact test, *t*-test for independent samples, and one-factor variance analysis were used to examine the association between the dependent variable (survival: no vs. yes) and qualitative and quantitative demographics, clinical, and intraoperative markers.

The Pearson or Spearman correlation coefficient, *t*-test for independent samples, and one-factor variance analysis were used to examine the association between the dependent variable (HHS and SPPB) and the quantitative and qualitative demographics in the one-dimensional analysis. All indicators with a *p*-value of 0.2 in the one-dimensional analysis were included in a multiple linear regression model using the entry technique to find the predictive indicators of functioning with the HHS and SPPB as dependent variables.

All of the prerequisites for the linear regression to be realized (homogeneity, linearity, normal distribution, and independence of the remainder of the model, as well as the linearity of the independent variables) were tested. To establish the prognostic indicators of survival, all indicators reported in the one-dimensional analysis with a *p*-value of 0.2 were incorporated in a multiple accounting regression model using the entry approach. The one-factor variance analysis model with iterative measurements for the temporal analysis of continuous variables was applied, and pairwise comparisons were performed using the Bonferroni test. All statistical analyses were carried out using the statistical software program SPSS VR 21.00. (IBM Corporation, Somers, NY, USA) and was performed by an independent statistician. All of the exams were two-sided. The *p*-value of 0.05 was defined as the level of statistical significance. 

## 4. Results

The study sample consisted of 100 patients over the age of 50, who suffered from a hip fracture. However, two patients were lost and eighteen died before the 12-month endpoint follow-up, thus 80 patients were finally included in the present study for further statistical analysis. 

The characteristics of the sample are shown in Table 1. The majority of patients were female (81.3%), with mean age of 82.9 years and mean BMI of 26.9 kg/m^2^. Of those, individuals reporting at least one fall over the previous year were 55%, and 76.3% of them had not undergone any previous orthopedic procedure. Approximately, 66.3% of patients had eyesight problems, and nearly half of them (46.3%) had used a walking aid in the past. The admission diagnosis was femoral neck fracture (45%), intertrochanteric fracture (45%), and subtrochanteric fracture (10%). The mean length of stay was 10.5 days and the majority (92.5%) completed the rehabilitation program. Finally, the average CCI was 6.9.

The TUG performance test, HHS, SPPB score, and Patient-Related Outcome Measures during the 12-month follow-up period are analytically described in Table 2. SF-36 general health, mental, social and functionality statuses marked a statistically significant drop from baseline. A statistically significant drop in EQ-5D scores was noticed in each of the subcategories, such as in mobility, self-care, daily activities, pain, stress, and the VAS index. The HHS recorded the greatest improvement between 6 and 12 weeks (52.1 ± 14.6), whereas the TUG score continued to improve significantly from 139.1 ± 52.6 s at the 6-week follow-up to 66.4 ± 54 s at the 48-week follow-up. The one-factor analysis of the SPPB in statistical significant relation with demographics and clinical characteristics is shown in Figure 1.

Table 3 presents the multivariate analysis of the HHS at 12 months using the multiple linear regression model adjusted for sex, age, BMI, falls during the follow up period, the walking aid use, the CCI, and the length of hospital stay. A one-year rise in age resulted in a 0.8 point decrease in the HHS index at 12 months, whereas women had an HHS that was 10.8 points lower than males. People who have had falls in the last 12 months had an HHS that was 7.7 points lower than those who had not experienced falls, and those who utilized a walking aid prior to the fracture had an 8-point-lower HHS than those who did not.

The multimodal analysis of the SPPB index at 12 months is shown in Table 3, using the multiple linear regression model, adjusted for sex, age, BMI, falls during the follow-up period, walking aid use, the CCI, and the length of hospital stay. A one-year rise in age led to a 0.1 point decrease in the SPPB index at 12 months, and women had a 0.92-point-lower SPPB than males at the same time. People who have had falls in the past 12 months had a 0.56-point-lower SPPB at 12 months following surgery than those who had no had falls, whereas people who utilized a walking aid before to the fracture had a 0.75-point-lower SPPB at 12 months following surgery than those who did not (Figure 2).

Survival at 12 months was strongly related with previous orthopedic operations and the use of a walking aid (Table 4). People who did not use a walking aid were 3.75 times more likely to die in the year following surgery than those who did. People who have had previous orthopedic lower limb surgery were 3.79 times more likely to die during the 12 months following surgery than those who had not previous surgery.

## 5. Discussion

Patients suffering from hip fractures usually have a history of physical frailty and reduced physical strength [30]. A series of measures are applied to test physical function, including mobility, endurance, muscular strength, and balance [31]. Following hip fracture, patients usually report a considerable decrease in HRQoL, particularly in self-care and daily activities [21]. Pain has been associated with a delay in walking during post-operative rehabilitation [32], making it a major component in patients’ capacity to exercise and subsequent recovery. Patients usually never regain their previous functional ability [33]. Literature data on possible determinants of physical function following hip fracture is limited. The SPPB has been shown to predict physical function up to 12 months following admission in previous cohorts of elderly patients discharged from acute care hospital settings [34,35].

The results of this study revealed that there is a statistically significant worsening of general health and its sub-sectors following hip fracture surgery at the 12-month follow-up compared to their pre-fracture state of health. The time analysis of the questionnaires SF-36 and EQ-5D/VAS indicates that patients are unable to regain the level of health they had before the fracture. Patients included in the study had a mean age of 83 years, and the average CCI was 6.9. These findings are in accordance with previous studies and show that despite significant improvements in patients’ health from discharge to the 12-month follow-up, they are unable to return to their pre-fracture levels of health. According to the existing information on restrictions in daily living activities, 29% of elderly patients with hip fracture do not regain their pre-fracture levels of mobility one year postoperatively [33]. Patients who successfully recover usually return to their previous levels of function within 6 months [33]. Their performance improves throughout the first year following fracture; however, the duration of recovery varies by functional area [36].

The time analysis of the HHS reveals a statistically significant difference in the HHS time estimations. Apart from the 24 and 48 week time estimates, parallel comparisons demonstrate a difference between all-time estimates. The greatest improvement in HHS values was seen up to 3 months after surgery, the smallest improvement was recorded between 3 and 6 months, and almost no improvement was observed between 6 and 12 months. The findings may be justified by the fact that the patients completed a three-month physical therapy program before being advised to continue the same program on their own for up to one year. Despite the improvement, the final scores remained low, indicating a poor functional outcome (HHS < 70). The one-factor analysis of the HHS at 12 months revealed that male patients with no falls in the previous 12 months before the fracture and those who had not used a walking aid had a higher HHS, which was marginally in the range of 70 (bad outcome) or between 70 and 80 (moderate outcome). Therefore, as age, body mass index, comorbidities, and inpatient care increases, HHS decreases. This is supported by Shyu et al., who indicated that patients required a longer time to reach previous levels of performance for more challenging and complicated tasks and that most of the functional recovery occurred in the first 6 months after hospital discharge [37]. Other studies suggested that rehabilitation is almost complete at the 6-month follow-up, and no significant improvement is documented between 6 months and 1 year [38,39].

The temporal analysis of SPPB reveals a statistically significant difference in SPPB time estimations. Apart from the 12- and 24-week follow-ups, pair comparisons revealed a very high significance difference between all-time estimates. There is no difference between the groups at 24 and 48 weeks. The data showed that patients’ balance, gait, and capacity to sit in a chair improved significantly from the time of discharge to 6 months. Nevertheless, the overall score at 6 months remained low, indicating that patients lacked independence and were at risk of falling. Furthermore, the slight improvement shown between 6 and 12 months was not statistically significant enough to change the clinical picture. According to the single-factor analysis of the SPPB index at 12 months with demographic and clinical indicators, ‘Male’ patients, those with no ‘previous falls in the last 12 months prior to the fracture’, patients who did not ‘use Walking Aid’, and patients ‘with a subcapital fracture’ had a higher SPPB score, with an average value greater than 4.5. Finally, the SPPB score at 12 months had a statistically significant connection with ‘age’, ‘BMI’, ‘CCI’, and ‘Hospital stay’. Therefore, as age, body mass index, comorbidities (measured by ‘CCI’), and in-hospital stay increased, the value of the SPPB decreased.

The SPPB score was proven to be an excellent indicator of global frailty in elderly people, and it appears to be a simple and reliable tool to identify persons of both genders who experience falls regularly. Low SPPB scores have also been linked to adverse outcomes, such as new falls, mobility loss, disability, hospital readmissions, prolonged length of stay, nursing home admission, and mortality [26,35]. In a cohort of 2710 cases, SPPB scores of 0–6 were substantially more related with the status of repeated fallers than SPPB scores of 10–12 [40]. The SPPB total score was linked to falls in a group of 307 cases [41] and 51 patients on hemodialysis [42]. The SPPB uses many measures to examine lower body function and has contributed mainly to establishing a relationship between lower body function and various health outcomes [43]. In hospitalized patients, low physical function is associated with poor outcomes. SPPB is also related to a lower risk of readmission and death in hospitalized individuals over the age of 75 [25].

The time analysis of the TUG reveals a statistically significant difference between TUG time estimations. Parallel analyses reveal a very significant difference between all-time estimates. The results show that patients’ walking speed improves significantly over the year. Nonetheless, at 12 months, walking speed remained high (mean = 66.44), indicating that patients do not have an independent mobility status and are at risk of falling. Nonetheless, the lowest number of falls (7.5%) was recorded in the first month following surgery, while the greatest (15.5) was observed between 6 and 12 months. This may be justified by the fact that in the first few weeks following hospital discharge, patients had limited mobility due to pain and muscle weakness. However, due to modern postoperative physiotherapy protocols, patients felt much sooner than they expected, leading to a more independent mobility before the surgeon’s or physiotherapist’s permission, thus having greater risk of falls. During the 12-month follow-up, 46.8% of patients experienced at least one fall. Participants who became recurrent fallers had substantially higher TUG times (71.6 s) at baseline than those who did not experienced falls (51.4 s) in the following 12 months, according to Wald et al. [44]. Nonetheless, earlier research has been inconclusive in terms of TUG’s capacity to differentiate between fallers and non-fallers, in part due to the wide range of selected cut-off points (varying from 9 to 32.5 s) [45]. The TUG performed at discharge with a cut-off point of 24 s was the only measure that substantially predicted declines over the 6-month follow-up period, according to similar results published by Kristensen [46].

The temporal analysis of walking aid usage following surgery reveals a statistically significant difference between the time estimations of the percentage of walking aid use. Except from the 1st and 6th weeks and the 24th and 48th weeks, the pair-wise comparisons demonstrated a difference between all-time estimates. Data showed that despite progressive recovery, 72.5% of patients continued to need a walking aid up to 12 months following surgery, although 46.3% of patients claimed that they were using a walking aid at the time of admission to the study. For a considerable proportion of patients, this suggested a failure to recover to the pre-fracture motor and functional condition, as well as an overall deterioration of health. The TUG, in hip fracture patients, is related to the type of walking aid. Patients conducting the TUG using a walker were on average 13.6 s quicker with a rollator. Patients who underwent the TUG with crutches were 3.5 s quicker using a rollator than with elbow crutches [47].

The current findings have implications for clinical management as well as future clinical trials, including hospitalized older adults. The study’s merits were that the SPPB score was independently associated to hip fracture rehabilitation status after adjusting for pre-fracture condition and mobility, and that the specific SPPB score for identifying probable predictive value for patients suffering from hip fracture was found. Second, the SPPB can predict clinical outcomes quickly, simply, and repeatedly without increasing expenses or taking up a lot of space, and it can be used to test patients’ functional capacity in outpatient and clinical settings securely. The findings of this study contribute to the growing body of data supporting regular clinical monitoring using the SPPB. Third, self-reported questionnaires are frequently used to assess physical function in older persons [48,49]. However, the adequate assessment of physical function in older persons with hip fracture may be problematic. Another advantage of our study is its population-based design and large sample size, which includes men and women who are typical of the overall community-dwelling older population.

The results of our study show that patients who did not use a walking aid before the fracture were 3.75 times more likely to die one year after surgery than those who used a walking aid. This result, although it seems paradoxical, is justified after a more careful observation of the data because a large part of the patients did not use a walking aid before the fracture, although they needed it. Overall, 57% of the patients answered the self-report questionnaires, when admitted to the study, that they did not use a walking aid before the fracture. However, 77.2% of them responded to the EQ5 mobility questionnaire that they had reduced mobility and did not have independent walking (perhaps walking with the help of a companion or holding on to walls and furniture). Overall, 53.2% of patients, who did not use a walking aid while having reduced mobility, reported a history of falls in the last twelve months before the fracture, with a mean age of 88 years and a mean CCI comorbidity score of 8, and 34.8% of them died within one year of the operation.

This study possesses certain limitations that should be noted when interpreting the results. The relationship between SPPB score and faller status does not always suggest that falls were caused by poor physical performance. Lower SPPB scores in fallers might potentially be the outcome of harmful falls or a more cautious attitude of patients toward mobility due to their fear of falling. As a result, our findings should be validated in population-based research involving both healthy and active people. Finally, several clinical factors, such as administered drugs and comorbidities, were not taken into account in the multivariate analysis. We couldn’t rule out the idea that the link between muscle mass and muscular strength, as well as physical performance, varies. Finally, the variability of fracture fixation methods utilized for the treatment of hip fractures may alter hip biomechanics, particularly during the acute postoperative period, and should be considered.

Prolonged postoperative rehabilitation can improve clinical outcomes and quality of life in patients with hip fracture. Specialized multidisciplinary rehabilitation management (physiotherapy and occupational therapy) contributes significantly to the improvement of the patients’ functional results (improvement of muscle strength, balance, and prevention of falls). It is important for orthopedists and physical therapists to adopt individualized rehabilitation approaches and to understand the strengths and weaknesses of various alternatives. Further research should focus on developing a stratified perioperative multidisciplinary hip fracture treatment and care protocol based on CCI and SPPB scores to determine the effectiveness of therapeutic interventions in improving patient functional outcomes.

## 6. Conclusions

This study proved that the SPPB was employed safely in a sample of elderly people. The SPPB is related with a lower incidence of readmission and death in individuals with hip fractures. The current study’s findings suggest that the SPPB can be used to achieve an accurate prognosis as a predictive tool for functional recovery in hospital settings. One of the primary purposes of acute geriatric units is to maintain functional abilities in elderly patients. As a result, the advantages of frequently monitoring the functional state of elderly patients might be considerable. The SPPB has been found to be a reliable method for assessing falls in elder people. As a result, this test should be used in clinical settings because it is easy to use, short, and standardized.

## Figures and Tables

**Figure 1 life-13-01748-f001:**
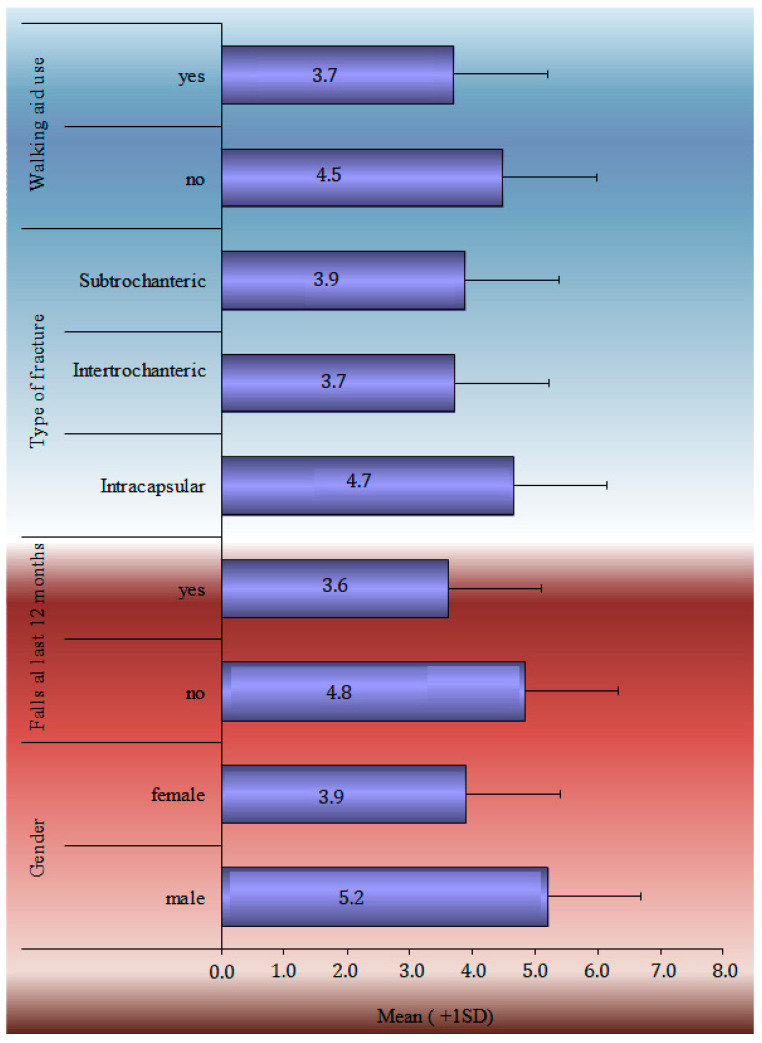
Univariate analyses of SPPB at 12-month follow-up in statistical significant relation with demographics and clinical characteristics.

**Figure 2 life-13-01748-f002:**
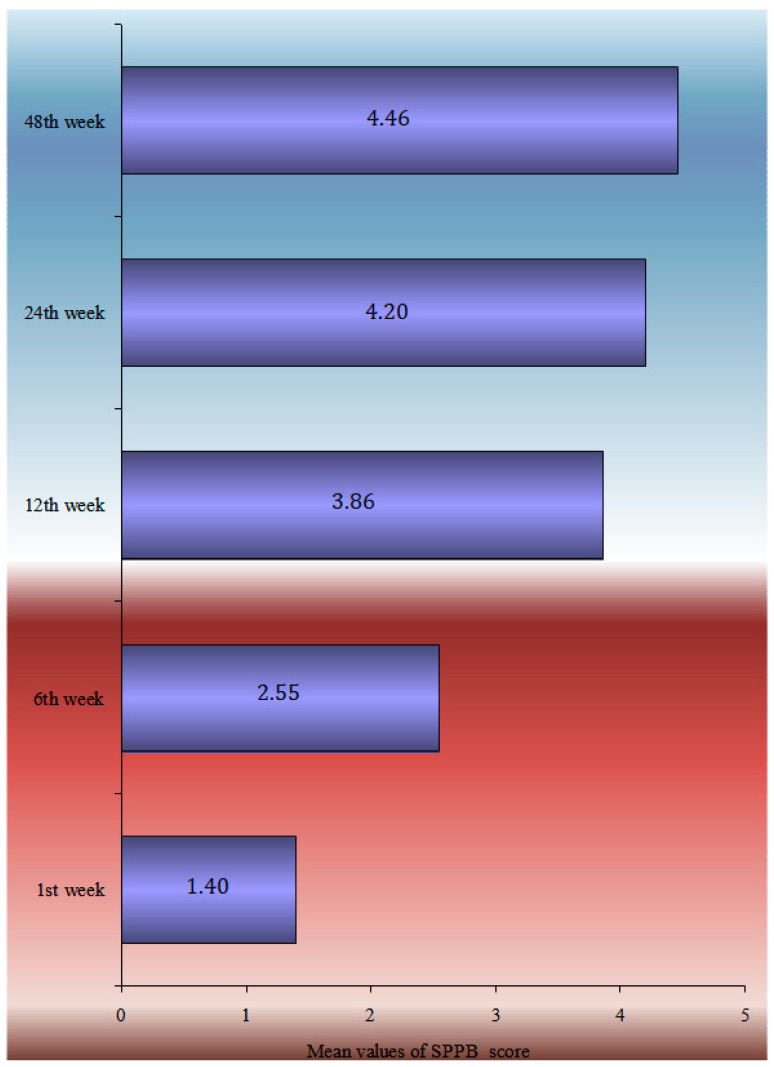
Mean parametrical SPPB change during the 12-month follow-up.

**Table 1 life-13-01748-t001:** Patients’ demographics and characteristics.

Demographics		Mean ± SD	Range	n	(%)
Age		82.96 ± 8.3	51–98	80	
Female				65	(81.3)
Length of stay		10.56 ± 3.88	5–23		
BMI		26.98 ± 4.52	18.80–46.31		
Falls over last 12 months				44	(55)
Previous orthopedic surgeries				19	(23.8)
Vision problems				53	(66.3)
Physiotherapy				74	(92.5)
Walking aid use				37	(46.3)
Diagnosis	Femoral head			36	(45)
	Intertrochanteric			36	(45)
	Subtrochanteric			8	(10)
Charlson Comorbidity Index		6.94 ± 2.20	2–13		

SD: Standard deviation, BMI: body mass index.

**Table 2 life-13-01748-t002:** TUG performance test, clinical function score with Harris Hip Score evaluation, Short Physical Performance Battery score and Patient-Related Outcome Measures during the 12-month follow-up period.

	Baseline	1 Week	6 Weeks	12 Weeks	24 Weeks	48 Weeks	*p*-Value
TUG			139.1 ± 52.6	104.5 ± 53	80.4 ± 51.8	66.4 ± 54	<0.0005
HHS		19.9 ± 9.9	36.7 ± 11.3	52.1 ± 14.6	59.2 ± 17.7	59.3 ± 20.6	<0.0005
SPPB		1.4 ± 1.3	2.6 ± 1.4	3.9 ± 1.5	4.2 ± 1.5	4.4 ± 2.1	<0.0005
EQ5D VAS	59.1 ± 18.7					47.1 ± 18.1	<0.0005
SF-36 functionality	29.7					15.3	<0.0005
SF-36 general health	53.1					29.2	<0.0005

Values expressed as mean ± SD, n (%), or median [interquartile range]. TUG: Timed Up and Go test, HHS: Harris Hip Score, SPPB: Short Physical Performance Battery score.

**Table 3 life-13-01748-t003:** Multivariate analyses of the SPPB and Harris Hip Score at the 12-month follow-up. Multivariate analyses: adjusted for age, sex, body mass index, falls during the last 12 months, walking aid use, type of fracture, length of hospital stay, and Charlson Comorbidity Index.

		SPPB					Harris Hip Score			
	R^2^	Reference Value	b	SE	*p*-Value	R^2^	Reference Value	b	SE	*p*-Value
Sex	2.9%	Male	−0.92	0.50	0.071	4.4%	Male	−10.82	5.2	0.041
Age	23.2%	-	−0.09	0.03	0.001	14.8%	-	−0.78	0.27	0.005
BMI	<0.5%	-	−0.01	0.05	0.807	<0.5%	-	−0.39	0.48	0.417
Falls	7.4%	No	−0.56	0.44	0.040	8.6%	No	−7.71	2.62	0.039
Walking aid use	3.8%	No	−0.75	0.41	0.075	4.2%	No	−8.04	4.31	0.066
Type of hip fracture	<0.5%	Subtrochanteric	0.40	0.40	0.319					
Charlson Comorbidity Index	2.2%	-	−0.16	0.11	0.129	<0.5%	-	−0.96	1.11	0.389
Length of hospital stay	<0.5%	-	0.01	0.05	0.797	<0.5%	-	0.43	0.56	0.443

**Table 4 life-13-01748-t004:** Multivariate analyses of survival at 12 months follow up. Multivariate analyses: adjusted for sex, age, BMI, prior orthopedic operations, walking aid use, neurodegenerative disease, physiotherapy program, Charlson Comorbidity Index, and length of hospital stay. Indicators with a *p*-value < 0.2 (in bold) were incorporated in a multiple accounting regression model.

	Reference Value	OR	95% CI		*p*-Value
Sex	Female	1.49	0.35	6.39	0.591
Age	-	1.07	0.98	1.17	**0.131**
BMI	-	0.92	0.79	1.08	0.307
Prior orthopedic operations (yes)	No	3.79	1.09	13.15	**0.036**
Walking aid use (no)	Yes	3.75	0.87	16.17	**0.076**
Neurodegenerative disease (yes)	No	2.43	0.55	10.62	0.240
Physiotherapy program (no)	Yes	1.69	0.29	9.89	0.558
Charlson Comorbidity Index	-	1.16	0.86	1.55	0.333
Length of hospital stay	-	1.11	0.95	1.31	**0.186**

## Data Availability

The data presented in this study are available on request from the corresponding author. The data are not publicly available due to privacy restrictions.

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
