# Peer review of "Rehabilitation Prognostic Factors following Hip Fractures Associated with Patient’s Pre-Fracture Mobility and Functional Ability: A Prospective Observation Study"

_life, 2023, doi:10.3390/life13081748_

Round 1

Reviewer 1 Report

Many thanks to the authors for having presented a so interesting original article about Rehabilitation Prognostic Factors following Hip Fractures associated with patient’s pre fracture mobility and functional ability. A prospective observation study.”.

Before resubmitting the revision version of the article, please read the editorial rules carefully, and check other editorial aspects, such as: text alignment (lacking), text justification at the head (lacking), etc. The language is no good, hence the manuscript needs to be corrected by a person of English mother tongue.

 Title and Abstract

The title cover the main aspect of the work.

The abstract is not well structured, although it contains the main results of the study, because the manuscript does not reflect the Strobe Statement-Checklist for cohort studies. Please read these guidelines for articles before resubmitting the revision version. Hence, make sure that the different paragraphs are divided properly. Please try to better describe the conclusion of the study.

 Key words

Please provide them in alphabetic order.

Background

The introduction is quite well structured, but it does not clearly identify the problem addressed in the study and does not clearly state the purpose of the manuscript. It is too short and some references are very old.  Pleas, rewrite this section.

Line 30: Half of the patients suffering from hip fracture regain independent gait and return to their previous mobility status and only 25% fully recover, while 20% may need long-term nursing care units [11,12].

Please, add more detail about hospitalization and quality of life of these patients, quoting:

·         Efficacy of an interdisciplinary pathway in a first level trauma center orthopaedic unit: A prospective study of a cohort of elderly patients with hip fractures. Arch Gerontol Geriatr. 2020 Jan-Feb;86:103957. doi: 10.1016/j.archger.2019.103957. Epub 2019 Oct 12.

Methods

This section contains enough information to understand and possibly repeat the study.

However, you should better explain which are the inclusion de exclusion criteria of the study.

 Statistical analysis

Describe the programs used for the statistical study.

Please provide who performed the analysis: an independent statistician or the same authors?

Results

The results presented are quite complete, reflecting the MM section.

 Discussion

The length and content of the discussion communicate the main information of the paper.

However, the article lacks more comparison with other studies in the literature.

Furthermore, it would be worthwhile to identify a model of action to improve post-operative control and the risk of falls in operated patients in the future.

 Conclusions

The conclusions reflect and refer to the results of the study.

 References

The references are not up to date. Hence, delate those before 2005 if not essential, replacing them with newer ones and that suggested previously.

Tables and Figures

The number and quality of figures and Tables are appropriate to transmit the main information of the paper.

Minor editing of English language required

Author Response

Thank you for your kind response and especially the reviewers with their helpful and precise comments. After thorough inspection of your comments, my co-authors and I proceeded with the appropriate changes made. Find attached the revised draft  (highlighted version).

We proceeded with some minor changes in English language throughout the manuscript. Keywords provided with alphabetic order, and some crucial sentences were added on the abstract and introduction section. Methods were enriched with inclusion/exclusion criteria and statistical analysis. Two rows were removed from Table 2, so as to make it much more interesting and less boring for the readers. Furthermore, we updated references, trying to be much more up-to-date, comparing to the previous manuscript submitted.

Reviewer 2 Report

The indroduction is laking of sufficient backround.

The novelty or the originality of the present study could not be justified by the presented data. SPPB has been already suggested as a reliable prognostic tool in the postoperative period of hip fractures and or geriatric patients.

Although you recorded the different surgical techniques used, you did not correlate SPPB with each of them seperately (the prognosis is not the same for each of them), making the specific data sample rather disparate.

Tables need considerable improvement. For example, table 2 should not include all these variebles. 

Lines 102-103: "Patients who died at any point thoughout the study were not included". Yet you do perform analysis of this individuals.

The quality of your English require some improvements.

Author Response

(The authors gave the same response as above.)

Round 2

Reviewer 1 Report

Manuscript has improved in this new version. However, the authors should have used MDPI template before resubmitting the modified version of their manuscript as suggested previously.

Minor editing of English language required.

Reviewer 2 Report

The authors proceeded with appropriate changes. I strongly believe that the paper could be published in the current form mostly due to its high quality of presentation and scientific soundness.

Minor editing of English language required.